# Barriers and Concerns in Providing Sex Education among Children with Intellectual Disabilities: Experiences from Malay Mothers

**DOI:** 10.3390/ijerph19031070

**Published:** 2022-01-19

**Authors:** Nawal Nabilah Kamaludin, Rosediani Muhamad, Zainab Mat Yudin, Rosnani Zakaria

**Affiliations:** 1Department of Family Medicine, School of Medical Sciences, Universiti Sains Malaysia, Kubang Kerian 16150, Malaysia; nawal_kamaludin@student.usm.my (N.N.K.); rosnani@usm.my (R.Z.); 2School of Dental Sciences, Universiti Sains Malaysia, Kubang Kerian 16150, Malaysia; drzainab@usm.my

**Keywords:** sex education, intellectual disabilities, barriers, concern

## Abstract

Though sex education (SE) may benefit the sexual development and overall well-being of children with intellectual disabilities (IDs), obstacles to its implementation remain. This study highlights barriers and concerns of SE for children with IDs based on their mothers’ experiences. We employed a phenomenological approach and in-depth interviews with twenty Malay mothers of children with mild-to-moderate IDs or/and other neurodevelopmental disorders. Four primary hurdles to SE were discovered: (1) mother (2) children (3) family value, and (4) socio-cultural environment. Inadequate knowledge, mothers’ perceptions that SE is less necessary at this stage of their children’s development, and time constraints were among their main barriers. Another source of hardship was the communication barrier because children with ID have cognitive impairment in their capacity to understand the topic being addressed and more time should be spent throughout the discussion. Family value and socio-cultural elements also had an impact on mothers’ intention to deliver SE to their children. Our findings suggest that mothers recognize the importance of SE for their children’s well-being. However, the dynamic interaction between the barriers complicates SE. This study emphasizes the necessity for future interventions to overcome hurdles at each level to effectively implement the recommended strategies.

## 1. Introduction

Many adolescents, particularly those with disabilities, get little or no formal sex and reproductive health education at school or home [1,2]. Yet, they need and receive benefits from education about sexual health, especially for their well-being [1]. The person with intellectual disabilities (IDs) may lack the decision-making ability, experience, and relevant skills required to create healthy relationships and appropriate sexual boundaries, which may result in sexual exploitation and other negative outcomes [3]. People with IDs often demonstrate a lack of knowledge about sexuality and limited access to sex education (SE), compared to their able-bodied peers [1]. They may believe the action or assistance by another person with their everyday activities such as bathing, dressing, touching, and socializing is acceptable. As a result, they are less capable of recognizing and reporting abusive situations and are more susceptible to manipulation by others.

There is a widespread misunderstanding among parents and the community that people with disabilities (PWDs) have no sexual needs because of their impairments and that SE is unnecessary for them [1,4]. This notion is very harmful to these children since they are more likely to be sexually abused and develop improper sexual behavior. As a consequence, PWDs have substantially less access to sexual health education and are at greater risk of sexual health issues owing to inadequate understanding of sexuality and abuse [1]. These socio-cultural norms have stymied research, public conversation, and acceptance of sexuality in PWDs [5]. SE has the potential to minimize these risks by empowering children and adolescents with disabilities and enhancing their capacity to seek assistance [6]. Comprehensive sex education (CSE) enables people with disabilities to have personal sexual satisfaction and defend themselves against abuse, sexually transmitted diseases, and unintended pregnancies [7].

Malaysia embraced a wide general definition of “learning disabilities” for registration reasons to obtain services and support from the government and agencies affiliated with the government. Under the Ministry of Women, Family, and Community Development, the Social Welfare Department (SWD) has developed seven categories of disabilities: speech, hearing, physical, vision, mental, learning, and multiple disabilities [8]. Individuals with learning disabilities have cognitive capacity that is not equivalent with their chronological age and also exhibit significant difficulties with their activities of daily life. These individuals include those with Down Syndrome, ADHD, autism, global developmental delay, intellectual disability, or specific learning disability such as dyslexia or slow learner. Upon registration, they receive a “Person with disability (PWD)” card that contains their personal information and type of disabilities. There is some stigma associated with “PWD” card holders and they are underserved and marginalized by the community [8].

The United Nations Educational, Scientific, and Cultural Organization (UNESCO) reported in 2015 that the majority of the children do not receive adequate and appropriate SE, leaving them uninformed when making decisions and vulnerable to abuse, sexually transmitted diseases (STDs), and unintended pregnancy [9]. The same scenario occurs in Malaysia where SE is a taboo subject in Malay society, despite it being crucial in preventing sexual abuse, sexual exploitation, and negative health outcomes against this group, which is particularly vulnerable [10,11]. UNESCO states that adolescents with disabilities need developmentally appropriate CSE to be safe and healthy and to achieve self-determination [9]. UNESCO defines CSE as “an age-appropriate, culturally relevant approach to teaching about sexuality and relationships by providing scientifically accurate, realistic, non-judgmental information” [9]. CSE aims to guarantee that young people receive comprehensive, life skills-based SE to acquire the knowledge and skills necessary to make conscious, healthy, and respectful relationships and sexuality choices [9].

There are compelling reasons for providing SE to children with IDs. PWDs have the same rights as able-bodied people in that they have the equal and similar potential to live a wholesome and fulfilling life. The Malaysian Government had made a step forward by enacting the Persons with Disabilities Act 2008 as a promotion of the betterment of their life. Under the section on promotion and development of the quality of life and wellbeing of PWDs, the PWDs have access to education and they shall not be excluded from the general education system to learn life and social development skills either by formal or by nonformal education [12]. Although Malaysia enacted the Persons with Disabilities Act 2008 (Act 685) and ratified the Convention on Rights of Persons with Disabilities (CRPD) in 2010, the Human Rights Commission of Malaysia (SUHAKAM) observes that a person with disabilities continues to confront inequality in many spheres of life [13].

Realizing the right to education and sexuality of children with disabilities are similar to those of the general population, the Malaysian government endeavored to provide SE to them especially since this group is vulnerable to sexual abuse as well as developing inappropriate sexual behavior. Joining and following the workshop and training session sponsored by WHO and UNICEF in 2004–2005 led to the development of the module “Live Life Stay Safe”. This module was designed to guide and assist parents, caregivers, and teachers in educating their special needs children or students regarding sexuality and SE, and anyone who provides day-to-day care for these youngsters. The content in this section may be used by teachers in conjunction with the existing special educational curriculum, or by health care practitioners during parent–child counseling or education sessions, and in the social welfare sector. This module covers both life skills (ensure the children with disabilities are capable of avoiding the situation and making a decision by their own choice) and personal safety (empower children with disabilities to maintain their right to be safe all the time) [14]. 

Apart from social and political factors, parents also play an important role in providing SE to their children with IDs. Stein et al. discovered most mothers acknowledged the value of SE, especially for their safety, despite having a negative attitude toward their children’s emerging sexuality, believing their children with IDs will not engage in sexual activity [15]. Gurol et al. also found all mothers in their study ignored SE for their children because they believed their children should not have a sex life [16]. When comparing with those mothers with able-bodied children, mothers of young people with IDs talked to their children about SE on a limited number of sexual topics and in less depth, addressed sexual matters at a later age, and the primary obstacle was children’s difficulty in comprehending [17]. Previous research has emphasized the difficulties parents experience when discussing sexual topics with their children with IDs due to a paucity of knowledge and training in SE, as well as children’s difficulty grasping the topic [17,18]. A previous review also showed that parental hurdles to delivering SE to their children with IDs are significantly influenced by family, religious, and sociocultural aspects [7,19,20]. 

Malaysian people, like those from any traditional nations, have a relatively scarce view of sex education due to cultural and religious traditions. As a consequence, the majority of parents do not openly discuss sex with their children or address matters connected to sex or body functions with their children. This was shown in research conducted among youth students with disabilities to explore their understanding about sex [21]. Surprisingly, none of the students said that their parents should be responsible for educating them about sex. Parents, in comparison to peers, seem less accommodating. Numerous examples demonstrate that parents did not discuss sex openly. According to some students, their parents often utilized kias (analogy) to discuss sex-related topics and used a variety of terminology in their discussion, making it more difficult to grasp. Similarly, according to studies, most Malaysian parents who were raised in culturally conservative environments have a negative view on their children’s sexual awareness [22].

The social ecological model (SEM) was adapted in this research to identify the obstacles that affected mothers’ attitudes when it came to providing SE to their children with IDs (Figure 1). McLeroy, Bibeau, Steckler, and Glanz (1988) developed a multilevel framework (SEM) based on the work of Urie Bronfenbrenner (1977) to articulate level-specific effects on health behavior and outlined potential intervention approaches at every level of influence [23]. This model was a well-known framework among policymakers for guiding public health practice [23]. This model reflects the dynamic interaction of individual, interpersonal, institutional, community, and social elements. This framework helped us to comprehend the range of barriers mothers encountered while delivering sex education. The model’s overlapping rings demonstrate how barriers at one level affect barriers at another. Future interventions will require an understanding at each obstacles level to ensure that the proposed strategies can be implemented successfully.

Understanding parental experiences and their views of their children’s needs relating to sexual health is essential to develop and implement relevant, effective, and successful interventions [10,24]. The limited research conducted in Malaysia to explore these issues was also highlighted [25]. Thus, this qualitative study aimed to identify obstacles and concerns mothers face while providing SE for children with IDs.

## 2. Materials and Methods

This qualitative study used in-depth interviews and a phenomenological strategy to elicit mothers’ perspectives about SE concerning their children with IDs. The qualitative approach enables the researcher to conduct in-depth explorations of the selected topics without being limited by predetermined categories of analysis and enables participants to express their perspectives [21]. By describing “what” and “how” mothers experienced in providing SE, the phenomenological method enables the researcher to dig into their views, perspectives, understandings, and emotions [22]. This research included mothers of adolescents (aged 10–19 years) with IDs and who were proficient in Malay. The Human Research Ethics Committee of USM approved the conduct of this study (USM/JEPeM/19080468).

### 2.1. Settings

This research was conducted in Kota Bharu, Kelantan, a suburban state in the northeast of Peninsular Malaysia. The main ethnic group is Kelantanese Malay. Kelantan’s population is 94% Malay, and all Malay are Muslims [26].

### 2.2. Participants

Mothers of adolescents with mild-to-moderate IDs and/or another neurodevelopmental disorder associated with IDs, such as attention deficit hyperactivity disorder (ADHD), autism spectrum disorder (ASD), or Down syndrome (DS), were recruited either from (1) the Child Psychiatry Clinic, Hospital Raja Perempuan Zainab II, Kota Bharu, or (2) from community-based rehabilitation (CBR). The recruitment strategies included using (1) key persons such as medical officers, teachers, or managers; (2) posters that we disseminated via a social-media group; (3) invitation letters. Those interested in participating in the study could either contact the key person or contact the main researcher directly. Once consented, the interview day and time were arranged according to the participant’s choice. The enrolment ceased after the data saturation was achieved.

### 2.3. Procedure

While waiting for ethical permission from the The Human Research Ethics Committee of Universiti Sains Malaysia (USM), researchers identify centers or organizations with an adolescent with intellectual disabilities. Following permission by The Study Human Research Ethics Committee, the researchers wrote a letter to the respected centers or organizations to seek permission to conduct research. 

After obtaining the approval from the center’s director or authorization, participants were identified based on inclusion criteria. Researchers received the contact number of the parents of an adolescent with disabilities from the teacher, manager, or other relevant person and scheduled an in-person interview with the parents. Parents who understood the research’s objective and volunteered to participate in the interview were selected and were given a participant information sheet containing detailed information about the research. On the day of the interview, participants gave consent before the session began. Following that, participants were requested to complete a sociodemographic questionnaire. After completing the form, the interview session started by using a semi-structured questionnaire to guide the researcher during the interview. This research was done in Kelantanese dialects to ensure that mothers felt comfortable sharing their perspectives and experiences. The question was open-ended, concise, and comprehensible. We started by asking participants the following important questions: “What is your view on SE for your children?”, “How do you offer SE for your children?”, and “What are the barriers you face?”, before moving on to more detailed and probing inquiries. The interview time limit was variable, although it was often about 60 min. The interview was held in a peaceful, comfortable, and distraction-free setting, such as at a children’s community-based rehab center, a hospital during follow-up, or anywhere the mothers chose. All interviews were audiotaped, kept in a secure location, transcribed into the text, and subsequently analyzed. Before the real interview, a pilot study was performed with three mothers and a senior co-researcher who was an expert in the qualitative study to evaluate the questionnaire’s acceptance, validity, and appropriateness.

### 2.4. Data Analysis

The audiotaped interview was transcribed verbatim. After that, it was coded using the NVivo (Qualitative Research Computer Analysis Package) software (Microsoft Corporation, Redmond, WA, USA). Thematic analysis was performed to extract relevant themes from the transcribed text for this research. Several strategies were used to ensure the rigor and credibility of the data analysis. To begin, the three researchers (NNK, RDM, and ZMY) carefully reviewed the first five transcripts to familiarize themselves with the parents’ general perspectives and views. Following that, the main researcher created an initial list of codes in NVivo^®^ (Microsoft Corporation, Redmond, WA, USA), and we coded the transcripts properly and consistently. We examined each transcript separately before collecting the identified themes and organizing them into an interconnected framework (themes, subthemes, and axial coding). To guarantee trustworthiness, reliability, and proper coding, the research supervisors (RDM and ZMY) collaborated with the main researcher to double-check all interview transcripts’ codes (NNK) to ensure confirmability. Any conceptual differences about thematic analysis were addressed and revised, and further preliminary themes were developed. Additionally, a co-researcher (RZ) with a background in children’s sexuality was recruited to examine and discuss the coding and early themes for all transcripts to offer general helpful feedback. In terms of reflexivity, none of the researchers had any prior contact with either the organization that aided data collection or any of the respondents. Finally, a consensus was reached on the final codes, as well as on the themes, subthemes, and axial coding. The transcribed material was sent to chosen participants to be reviewed to verify the results and maximize the credibility. Various alternative interpretations were addressed. None of them had a contrary view to these results.

## 3. Results

### 3.1. Characteristics of the Participants

This study involved twenty Malay mothers of adolescents with IDs. Twenty-three eligible mothers were called; however, three were unable to participate owing to their busy schedules. The characteristics of the mothers who participated are summarized in Table 1. The mothers’ mean age was 48.8 years. They were mostly married. Over half have a secondary education and were homemakers. The median age of the children was 15.4 years, with an equal representation of adolescents in their early, middle, and late adolescent years. In terms of disability, about one-fourth have IDs or DS, one-fifth have ASD, and about one-third have mixed diagnoses.

### 3.2. Themes

Two main themes developed as a result of the mothers’ experiences: hurdles in providing sex education and mothers’ concerns and needs (Table 2).

### 3.3. Theme 1: Hurdles in Providing Sex Education

We explored the challenges mothers encountered while educating their children with IDs about sexuality. Four significant hurdles to delivering SE have been identified: (1) paucity of knowledge impedes the role, (2) perceived SE is less necessary, (3) communication barrier, and (4) family value and *Adat* (culture-norm).

### 3.4. A Paucity of Knowledge Impedes the Role

All mothers with a low-to-moderate level of education expressed that they have a lack of knowledge or skills in delivering SE to their children with IDs, which may impede their ability to successfully fulfil their role as the main sex educator. Mothers mentioned that they were not previously exposed to adequate SE in their life, rendering them unable to effectively educate their children.

According to P6, she educated her child about sexuality based on her self-understanding. She received and learned some sexual knowledge previously from her elders during child age, or during her previous school years, or via personal experiences. However, she confessed that the information she gave to her child was rudimentary and not detailed. *“I don’t mind talking about it…but sometimes my knowledge hinders me to do so…I just shared within my context knowledge…the basic ones I would say.”*

Another mother, P3, said she was unsure where to begin or what to discuss with her son about sexuality. She lacked the expertise and ability to educate her son on SE since she had never addressed these topics with her previous children. For her, starting a conversation about sexuality is not an easy task.


*“I’d rather say I have never talked about this sexual issue with my children…as I don’t know how to start the conversation… This issue offers me a great concern…but I’m embarrassed to bring it up… I’m clueless to put it into appropriate utterances. I don’t know how to put it into appropriate words to make it brief.”*


### 3.5. Perceived Sex Education Is Less Necessary

All the mothers in this study realized the importance of SE for their children with IDs; however, they believed the SE that their children received should be age-appropriate, based on their children’s comprehension level, their mental capacity to comprehend the discussed topic, and not in-depth discussions that involve the topic of sexual activity. Moreover, the majority of mothers in this study believed that SE is unneeded at their current child’s age for a variety of reasons.

#### 3.5.1. Limited Socialization Activities

Many mothers believed that their children with IDs did not need SE at this stage in their development. SE, according to some mothers, was superfluous given their children’s restricted social engagements. They have restricted social interaction with the rest of the community after school, which minimizes their children’s susceptibility. P5 stated that her child’s life had been confined to the house and that everything she did was under parental control. *“I believe my daughter could live on her own (know how to take care herself) when I’m no longer around. She has been an easy one…prefers to stay at home…even if she insisted, she will be around just with the neighborhood”*.

Similarly, P11 stated that her child’s impairment hampered his social interactions and prevented him from being sexually harassed. *“My child has always been easy so far, so I don’t talk about it. He’s well-behaved. Rarely hanging out either. He too puts me at ease always. I won’t worry much, not even of the likelihood over sexual harassment”*.

Few mothers believed their children were at low risk of being manipulated. The high level of trust among family members, as well as the home environment, renders sexual exploitation of the children implausible.

#### 3.5.2. Encourage Children’s Curiosity

Two educated mothers even perceived that SE would increase sexual practice because their children will misinterpret the discussed topic. P1 cautioned while discussing sexuality with her son and avoided lengthy discussions to prevent misunderstanding of the issue unless specifically asked by her son.


*“I would never go into details… its after-effect worries me the most…he could get so obsessed over girls…so it’s better don’t…An autistic child couldn’t control his obsessions once it grows...hence, I wouldn’t dare to [talk about it]. It will lead to misinterpretation. It’s risky, to be frank enough…an autistic child would practically ask for further details like an expert, it might be beyond our expectations.”*


Similarly, P4 expressed fear that the conversation topic would lead to her son’s exploration or practice. *“That’s one of my biggest concerns…it is different when the discussed subject is about the regular basis topic such as on a car…or perhaps its accessories, for instance, they insist to see it, even demand to experience it by themselves”*.

### 3.6. Communication Barrier

Mothers to children with ID had not only have their children’s mental capability limitations that challenge communication about sex education to contend with, but they also had their own time barrier.

#### 3.6.1. Delayed Mental Capacity to Understand

Children with IDs had a deficit in their ability to comprehend. Discussing the sexuality topic with their children with IDs was tough for the majority of mothers. Mothers were concerned about their children’s incapacity to absorb knowledge in the manner in which he or she should have, which will increase the likelihood of misunderstandings and misinterpretations of the discussion subject, which may result in improper sexual behavior. 

All mothers told us that their children’s cognitive ability and mental capability to comprehend the topic under discussion were far below the level for their actual age. P1 told us the primary challenge in conversation was explaining the matter in simple, and understandable language.


*“Any explanation needs to be elaborated accordingly, conforming with one’s cognitive ability…for instance, as for someone standard 6 (12-year-old), I’d say his cognitive ability would probably the same as standard 3 (7-year-old) children…these children perceive rules as it is…they are not keen to develop on how’s and why’s of such rules are referred as proclaim…there is no such rigid indicator to acknowledge the depth of their feelings too…their vocabulary correspondingly plays a crucial role. It takes a village to emphasize such feelings if they did not acknowledge it in the first place.”*


Similarly, P9 made a similar remark about her son’s mental ability to comprehend the matter being addressed, which might hamper the conversation. *“I’d say he’s cognitively (interpreting information) at the age of 9- or 10-year-old boy regardless he is currently 16.”*

Despite the communication gap between mother and child, P1 is enthusiastic to educate her child according to his maturational stage. *“By any chance, I would eventually talk about the sexual issue when its time which means it is acceptable of their maturity level accordingly…I would explain about it conforming with their development…at the right time under a certain appropriate condition I must say.”*

To overcome the communication barrier, P2 advocated using visual mediums to promote communication and learning in children with disabilities rather than only auditory input, as she did with her son. She should also assess her son’s comprehension after each explanation. 


*“He prefers to be explained through visual…or a video…that’s why we couldn’t talk much with him…he’s more into visuals…I have been integrating visuals to introduce something new for him ever since…I won’t deny it is somehow quite challenging to cope with the special needs child. A normal child can comprehend well within a standard timeframe, and they won’t hesitate to ask when the need arises…contrarily with the autistic child, there is a compulsory demand for me to double-check whether or not he understood on a particular area…”*


#### 3.6.2. Time Constraint

Other communication barriers involved mothers’ factors. Some mothers said that dealing with sexuality discussions takes a lengthy time and large effort, which limits their commitment to communicate effectively about the topic. P8, for example, stated that she spends less time at home, which limits her ability to be a main sexual educator. Rather, she contributes at a community rehabilitation center as a sexual educator for children with disabilities. *“I’d rather say I don’t expose my children to sex education as much as the outsiders. It is probably due to the time constraint I would say. I spend approximately 60 to 70% of my day at work whilst the remaining, I shall be at home”.*


### 3.7. Family Values and Adat (Culture-Norm) 

Some mothers felt that family values and culture-norms influence them so much in making a decision in providing SE to their children with IDs. P3 and P12 stated that although the family considered sexuality as a nonsensitive matter, discussing it openly remained a taboo topic and was not welcomed. *“How do I say…I think it (sexuality) would be a very sensitive topic frankly speaking…so I’d rather let it slide’*. *‘I think it would be a very sensitive topic, but it solely depends on an individual. I’m all ears to openly discuss it with family members. Otherwise, I might keep it silent”.*

P2 agreed that sexuality is a sensitive topic in Malay culture that is never discussed freely in the family. Additionally, some people used euphemisms to refer to reproductive organs, which is permissible in Malaysian society.


*“Haa… I doubt this topic has been a quite sensitive one even in our adat, our parents too have always been constantly believed in such insights… even if we were to address the reproductive organs such as a vagina or a testis, we will replace them with another word…we will not address them directly”.*


In contrast, P15 said that her extended family supports her efforts to educate her children about sexuality. For example, her aunt advised her daughter to dress properly when she went outside to avoid being exploited by others. She could also speak freely and have open discussions with her other family members if she wants to get more information or opinion regarding her daughter’s sexuality.

P9 also expressed a similar stance, stating that sexuality cannot be a taboo subject in the modern-day. *“It is not a sensitive issue…it shouldn’t be mindlessly discarded since it is compulsory for both parties; ours and our children to raise such awareness. We must make haste on the exposure of the sex education.”*

### 3.8. Theme 2: Mothers’ Concerns and Needs 

#### 3.8.1. Becoming a Vulnerable Group 

Seventeen mothers expressed concern that their children will become vulnerable to sexual abuse, manipulation, or exploitation when approaching adulthood as a result of their cognitive impairment, which may impair their judgment toward others, as portrayed by P16. *“Their innocence has caused these special needs children to be of among the sex abuse and rape victims…they would perceive any physical contact with strangers as a normal gesture hence they will not spontaneously reflect on the inappropriate action.”*

Such concerns were also raised by P6 as the perception that a child will be manipulated or exploited by other people because of their naïve and low level of understanding. This can be demonstrated by the following:


*“My concern lies on sexual intercourse …I will not be around all the time, to monitor her closely…I could not predict what would happen to her if she’s out of my sight…a sexual attraction between opposite genders… she will stay clueless…I’m worried over one’s intention towards her due to her impaired cognitive ability.”*


On the other side, P16 and P4 expressed worry that their children may become sexual harassers without intending to do so and may face repercussions from both the community (verbal and physical attack or ostracism) and the law (filing a police complaint against them/conviction). P16 said,


*“I have a daughter and it reciprocates on both parties…as we couldn’t control one’s desires…I have once heard previously that their desires (the special needs children) are more than the normal children…hence it is a wake-up call for me…it will be a mess too as I have eldest daughters…by any chance, his desire could have taken over his soul while others were sleeping soundly…to be frank, my thoughts incline more on the unpleasant experience”.*


Similarly, P4 raised her anxiety: *“I’m worried over the likelihood of him practicing so to other girls. His desires would have taken over his consciousness on his feelings… its after-effect doesn’t bother him anyway. I’m worried he will invade the opposite gender’s privacy. Surely others will misinterpret these special needs children’s actions.”*

Contrarily, two mothers did not foresee their children’s vulnerability to social manipulation or improper sexual behavior owing to their children’s limited social life, as mentioned before, and what they did under parental supervision.

#### 3.8.2. Need Proper Sex Education

We asked the mothers regarding their motivations for providing SE to their children despite all the obstacles. All the mothers admitted that SE is very important for their children as sexual development is growing along with the children’s age. Furthermore, SE can help them to have an independent life especially when their parents have left the world. P2 stated that SE is crucial since puberty and maturity are part of their children’s developmental stages. *“The physical growth and sexual maturity level are gradually developing regardless of the cognitively impaired. Hence, they should have been exposed as much as the normal children…the needs of doing so should be fully acknowledged by parents.”*


Mothers expressed concern about their child’s life after they passed away. Most parents hope their child will live independently and stay in a safe and healthy environment. P7 hopes that CSE can help her son can live independently when she is no longer around. She hopes her son can stand with his own two feet when he grows up.


*“Such concerns sometimes keep bothering me. How can my kid live on his own when I’m no longer around? It will be onerous to seek for people’s help even if it was his granny or perhaps his uncle when each of them has got too much on their plates too.”*


In addition, P16 wished CSE will provide a safe and secure future for her child when she is no longer around.


*“I regularly have an adult conversation with my kid…we engage in such conversations… It is worth knowing for him an early exposure as I too am growing older…I shall leave a room for him to learn before it is too late…that is what I have been emphasizing…come to think of it, it is inevitably an emotional moment to be reminded of…but I have to.”*


We asked mothers to provide recommendations about SE for their special needs children at the end of the interview. The majority of mothers expressed gratitude for the collaboration of several parties, such as nongovernmental organizations (NGOs), schools, and health care providers in delivering sex education in the future. P15 remarked,


*“I’d like to participate in any talks or programs on sex education or maybe a discussion on communicative competence between parents and a child…for instance, a program initiated for mothers with specials needs children conducted by the NGOs or perhaps by the medical associations, I’m keen for it… I could gain new experiences throughout the sharing session…and I might as well share with others.”*


## 4. Discussion

Even though the sexuality of children with IDs is already a sensitive subject that is seen adversely by the community, IDs add another degree of difficulty. IDs may make it more difficult for children to acquire sexual knowledge, owing to a lack of mental capability or a chance to address the sexual issues with friends, as well as a lack of cognitive abilities to search independently via online or offline platforms [27]. This study identified four main obstacles to SE for children with IDs: (1) mother, (2) children, (3) family dynamic, and (4) socio-cultural environment. The paucity of knowledge, lack of skills, perception that sex education is less necessary at the children’s development stage, and time constraints are the primary obstacles that mothers encountered while providing SE to their children with ID. Other difficult challenges for mothers were children’s communication barriers; children with IDs are deficient in cognitive ability to comprehend the discussed subject. Family support and socio-cultural factors also influence the mothers’ possibility to provide SE to their children with IDs.

In our study, all the mothers with a low-to-moderate level of education expressed that they lack enlightenment or experience for delivering SE to their children with IDs, which may hinder their capacity to effectively fulfill their role as the primary sexuality educator. This research discovered that parents’ level of sexual knowledge and attitudes concerning SE are inextricably linked to their educational level. In general, the greater the academic level of the parents, the greater the extent of sexual knowledge and the more favorable the attitudes toward providing SE to their children. This is in line with study by Xin Jin among Chinese parents that showed parents’ educational background had a significant effect on parental sexual knowledge and attitude [28]. As a result, parents with low academic level had a negative attitude towards SE due to scarcity of knowledge. Inadequate knowledge and previous skills often led to embarrassment when discussing sexuality [2]. This finding is consistent with many prior studies demonstrating this obstacle to parents in providing effective SE to children with IDs, where knowledge is the key for successful discussion [18,29]. As a consequence, the SE given to their children with IDs was insufficient [2]. 

Parents also revealed that a paucity of knowledge in SE contributes to their reluctance and passivity when it comes to providing SE to their children with IDs [30]. Similar findings were also observed in the previous study, which discovered that lack of knowledge and experience happened not just among parents, but also among sexual educators, such as a teacher. Ang and Lee also said that a lack of expertise has caused educators to be inept when it comes to SE [31,32]. This became one of the shortcomings encountered by sexual educators in school in implementing CSE to the students with special needs [31]. Thus, increasing awareness and accessibility on sex education is a very effective strategy for improving sexual knowledge and attitudes about sex education among parents, educators, and the general public.

Mothers believed that SE was less necessary at this child’s age as they considered their children with IDs lacked sexual feelings or needs [1]. Moreover, their children had not engaged in any improper sexual behavior, and their lives were under parental supervision. Numerous mothers expressed worry that SE might heighten sexual desire or activity, which could lead to inappropriate sexual behavior, similar to previous studies [19,33]. In contrast, many reviews of the literature have stated that SE may assist in the sexual development and sexual demand of children and young PWD, as well as contribute to their health and well-being [2,34]. Parental misunderstandings about their child’s sexuality are one of the obstacles to overcome [1]. Parents must arm themselves with an understanding of sexuality to be more ready and competent to educate their children with IDs. 

It was challenging for all mothers to explain to their children with IDs the physical, emotional, and social elements of sexuality. IDs are the intersection between an impairment that limits a person’s cognitive and adaptive function [27]. Parents stated that children with IDs had a limited understanding and mental capacity to comprehend the topic being addressed, which is far below that normal for their chronological age, in line with the previous studies that reported children’s comprehension is the main barrier in providing SE [18,19,30]. This barrier may influence a mothers’ attitude toward providing education for their children either favorably or negatively. Some mothers viewed it as a challenge and persisted to discuss it with their children until they understood. However, some mothers saw it as a hard challenge and expressed little interest in participating in SE [35]. However, there is room for improvement. Katz and Ponce suggest that while providing SE, it is crucial to use language that is comprehensible by the children [27]. Children with disabilities must be taught via a variety of mediums to have a thorough understanding of the topic being addressed. Parents should be creative when educating their children with IDs about sexuality.

We discovered in our study that family value and socio-cultural factors are significant barriers in the mothers’ preference to offer SE to their children with IDs, comparable to a previous study in which discussion regarding sexuality that is open and comprehensive may be hampered by discomfort caused by the environment factor, society, culture, or religion [7,19,20,36]. According to Pryde and Jahoda’s study, among Muslims, sexuality is a culturally sensitive topic that is rarely discussed freely inside the family [19]. Conversely, in a study by Ariadni et al., sexuality is not a taboo topic for mothers, and they incorporate religion to educate their children about SE since it is much more convenient. Our findings showed that mothers had a negative attitude toward providing SE to their children with ID when they did so selectively and not comprehensively. They were concerned that SE will increase sexual behavior and exploration in their children, despite the reality that all of the mothers in this study acknowledged the importance of SE for their children. A similar finding in the literature review found that parents are more worried about sexual abuse and unintended pregnancy than with the overall development of their children as sexual beings. As a result, parents’ views lead to insufficient SE that focuses only on relationship restrictions [1].

At the closing of the interview, mothers expressed some of their recommendations for stakeholders to consider facilitating them in overcoming these obstacles. Mothers felt that if all authorities cooperate, a structured, nondiscriminatory CSE can be given. One of the suggestions made by all the mothers in this study is for the school to include parental involvement in the SE curriculum for PWD. Parental cooperation may facilitate the delivery of SE to children with IDs from home to school. Previous research has shown that implementing SE to students with disabilities is difficult to accomplish without active or passive parental participation [37,38]. According to Chai Tin and Lay Wah (2019), this scenario is different in Malaysia, where teachers stick to the ministry of education guidelines based on the current curriculum without engaging parents [32]. Among the initiatives that the ministry of education may implement in the future is to engage parents in the SE curriculum. Another option accessible to relevant authorities such as the Ministry of Women, Family, and Community Development or nongovernmental organizations (NGOs) is to organize more forums for SE, either online or offline platforms, to address these topics with parents.

We acknowledge the limitations of our study and have recommendations for further research. Due to the small sample size, which included only mothers, and the narrow sociodemographic distribution to suburban areas, the results of this study cannot be considered representative of all parents of children with IDs. Further research might include both mothers and fathers to elicit their perspectives on these issues and widen the study area to include urban and suburban parents to see if there are any differences in the parents’ experiences and views. Future studies may potentially combine qualitative and quantitative methods to provide a more thorough outcome. There are several justifiable explanations for the study’s shortcomings, including the fact that despite efforts to recruit mothers and fathers from different ethnic backgrounds, the effort was unsuccessful. The research was conducted during the COVID-19 pandemic that restrict the participant’s recruitment, and the mother is the main caregiver for the majority of children in Kelantan. Additionally, the bulk of Kelantanese people are Malay.

## 5. Conclusions

Our findings suggest that all the mothers recognize the importance of SE for their children with IDs, particularly to prevent sexual abuse, sexually inappropriate behavior, or unfavorable health outcomes such as pregnancy. However, the dynamic interaction between the barriers, which include personal factors, child’s disabilities, cultural misconceptions (of being asexual), and a sensitive issue (sexuality), complicates the process of providing SE. Lack of knowledge and competence, the diversity of children’s disabilities, and the influence of social-cultural values in daily life contribute to the complexity of an already challenging situation to the mothers. This study emphasizes the need for future interventions to overcome the barriers at each obstacles level to ensure that the proposed strategies can be implemented successfully. It should be a collaborative effort including stakeholders, health care practitioners, school educators, and social care services to overcome the obstacles and support parents in delivering SE to their children with IDs.

## Figures and Tables

**Figure 1 ijerph-19-01070-f001:**
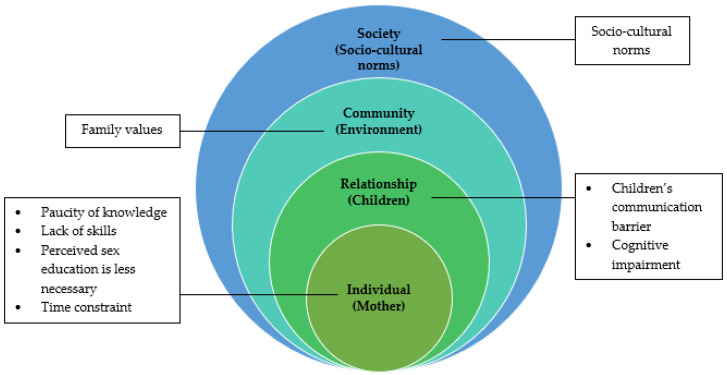
Adapted theoretical framework from McLeroy, Bibeau, Steckler, and Glanz Social-Ecological Model.

**Table 1 ijerph-19-01070-t001:** Characteristics of the mothers and children with intellectual disabilities (n = 20).

Identification	Age	Status	Education Level	Occupation	Child’s Diagnosis	Child’s Age	Child’s Sex	No. of Children
ID 1	44	Married	Master	Lecturer	ASD	12	M	4/5
ID 2	45	Married	PhD	Lecturer	ADHD with ASD	12	M	3/5
ID 3	59	Married	Secondary school	Homemaker	Mild CP with ID	15	M	4/4
ID 4	39	Married	PhD	Lecturer	Mild ASD	12	M	2/4
ID 5	39	Married	Primary school	Homemaker	Moderate ID	15	F	2/3
ID 6	37	Married	Secondary school	Homemaker	Mild ID	10	F	2/3
ID 7	43	Married	Secondary school	Medical Attendant	Mild ID with ASD	16	F	
ID 8	46	Married	Diploma	Staff Nurse	Mild ID	17	M	¼
ID 9	40	Married	Degree	Teacher	Autism	16	M	1/7
ID 10	60	Widow	Secondary school	Homemaker	Mild ID	19	M	1/7
ID 11	60	Married	Primary school	Homemaker	DS	18	M	7/7
ID 12	57	Widow	Primary school	Homemaker	DS	16	M	2 /2
ID 13	56	Married	Secondary school	Homemaker	DS	14	M	7/7
ID 14	48	Divorced	Secondary school	Cleaner Supervisor	Mild ID with hearing problem	18	F	3/5
ID 15	38	Married	Secondary school	Cleaner	DS	15	F	2/3
ID 16	52	Widow	Secondary school	Homemaker	ASD	19	M	¾
ID 17	55	Widow	Secondary school	Traditional massager	Mild ID	18	F	3/3
ID 18	58	Widow	Secondary school	Homemaker	DS with ASD	18	M	3/3
ID 19	41	Married	Secondary school	Homemaker	ASD with ADHD	14	M	2/3
ID 20	58	Married	Diploma	Retired (Ex-chief nurse)	DS	13	M	6/6

**Table 2 ijerph-19-01070-t002:** Barriers and concerns in providing sex education among Malay mothers, (n = 20).

Themes	Subthemes	Axial Coding
Hurdles in providing sex education (SE)	A paucity of knowledge impedes the role	Lack of knowledge and skills in delivering SE Not exposed to adequate SE in life previously Incapable of becoming an effective primary sex educator
	Perceived sex education (SE) is less necessary	Child with ID has restricted social life Providing SE is not a usual task even with able-bodied children The task of teachers in the school Children’s life under parental supervision Encourage children’s curiosity in sex
	Communication barrier	Difficulty in comprehending SE due to delayed mental capacity to absorb the information Need to use another medium such as visual and audio in delivering SE Time constraint- work schedule reduces time at home and takes a longer time for explanation
	Family values and adat (culture-norm)	Taboo topic for open discussion in conservative family In open minded family, sex discussion is discussable Depending on the level of sensitivity of the topic
Mothers’ concerns and needs	Becoming a vulnerable group	Easily to be manipulated or exploited by others, difficulties to differentiate between right or wrong Sexual abuse, become sexual harasser.
	Need proper sex education	Sexual development is growing with children’s age Child will live in safe environment by gaining the knowledge Live independently in future

## Data Availability

Data are contained within the articles.

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
