# Peer review of "Barriers and Concerns in Providing Sex Education among Children with Intellectual Disabilities: Experiences from Malay Mothers"

_ijerph, 2022, doi:10.3390/ijerph19031070_

Round 1
Reviewer 1 Report
Using qualitative in-depth interviews with Malay mothers who have children with mild to moderate intellectual disabilities, this study documented hurdles to sex education provided by the mothers and the mothers’ concerns and needs. While this qualitative study has potential for publication in peer-reviewed journals, revisions are needed.
First, to prepare the reader, I suggest that the authors develop a paragraph in the introduction section to describe social and cultural contexts in Malaysia that are relevant to sex education. For example, who typically provides sex education, in what settings (e.g., schools, homes), and unique challenges faced by children with intellectual disabilities in Malaysia.
Second, it would be very important for your readers to understand sex education in terms of the Islamic doctrines and religious traditions, which can be used to highlight the need for sex education among children who are intellectually disabled.
Third, the theoretical framework was appropriate but misplaced. Instead of introducing the theoretical framework in the discussion section, I would develop a literature review section aft the Introduction. In so doing, this theoretical framework can be used as a guidance for data analysis and directions for future research.
In the discussion and conclusion sections, I would like to see more detailed recommendations in terms of possible evidence-based intervention strategies and programs that are appropriate for sex education provided for adolescents with IDs in Malaysia. How would such programs affect and be affected by public policies in an Islamic society?
Author Response
Reviewer Comments
|
Comment |
Response |
1 |
Reviewer 1 Using qualitative in-depth interviews with Malay mothers who have children with mild to moderate intellectual disabilities, this study documented hurdles to sex education provided by the mothers and the mothers’ concerns and needs. While this qualitative study has potential for publication in peer-reviewed journals, revisions are needed.
|
Thank you. We appreciated reviewers’ comments and will revise accordingly. Thank you for the opportunity given. |
|
First, to prepare the reader, I suggest that the authors develop a paragraph in the introduction section to describe social and cultural contexts in Malaysia that are relevant to sex education. For example, who typically provides sex education, in what settings (e.g., schools, homes), and unique challenges faced by children with intellectual disabilities in Malaysia.
|
Thanks for the recommendation. Unfortunately, I can’t download the Fida Sanjakdar work freely, so we use research work from our country.
In the introduction: page 2-3: In Malaysia, SE is informally taught to Malay children in terms of appropriate personality trait and desirable behavior – to behave in ‘feminine’ way for girls and ‘masculine’ ways for boys around four years of age [12]. They are also taught that there are topics that cannot be talked about, including sexual intercourse, private body parts and sexually bodily functions [13]. Islamic teachers teach SE when the children approaching teenage years, mainly on menstruation issues, dress code, and avoiding immoral activities or illegal conducts topics as deemed by the law of the country and social norm of the community. When the person approaching end of teenage years or prior to marriage, roles and rights of married couple are taught in a formal workshop classes (Pre-marriage courses [13]. This is made possible by the law that requires all person applying for marriage to have the certificate of having completing this course [14].
Formal SE is not easily implemented in Malaysian School Curriculum. The government started to integrated SE into national curriculum for secondary school in 1989 while for primary schools, it was started five years later. The SE module was integrated in various academicals subjects such as in human science, Islamic education, languages and moral education [15]. However, it implementation creates debates among the society up to the scholars’ level, partly due to the incomplete training of the teachers who are supposed to handle the topics [16]. After taking into consideration about the sensitivity of the topic in the community, SE module has been renamed to become Reproductive Health and Social Education (RHSE) or Pendidikan Kesihatan Reproduktif dan Sosial (PEERS) in 2011 and delivered as a part of the Health Education subject [17]. Despite of all these effort, reported misconduct among school children with IDs still persist. Lack of understanding among the parents about the implementation of SE at school was found to be one of the reason causing low effectiveness of the program to educate special needs children in this matter [16].
In the introduction: page 4 we add: Malaysian like any traditional nations, have a relatively scarce view of sex education mostly due to cultural traditions and partial understanding of the religious tradi-tions. As a consequence, the majority of parents do not openly discuss sex with their children or address matters connected to sex or body functions with their children. This was shown in research conducted among youth students with disabilities to explore their understanding about sex [27]. Surprisingly, none of the students said that their parents should be responsible for educating them about sex. Parents, in comparison to peers, seem less accommodating. Numerous examples demonstrate that parents did not discuss sex openly. According to some students, their parents often utilised analogy to discuss sex-related topics and used a variety of terminology in their discussion, making it more difficult to grasp. Similarly, according to studies, most of Malaysian parents who were raised in culturally conservative environments have a negative view on their children's sexual awareness [28].
In the conclusion: Page 12
References added: 12. Muhamad, R.; Horey, D.; Liamputtong, P.; Low, W.Y.; Sidi, H. Meanings of Sexuality: Views from Malay women with sexual dysfunction. Arch Sex Behav. 2019, 48, 935–947. 13. Morni, A.; Johari, A.; Ahmad, J.; Jusoff, K. The linguistic taboo between Malays and Ibans of Sarawak, Malaysia. Canadian Social Science. 2009, 5, 141–158. 14. Saidon, R.; Ishak, A. H.; Alias, B.; Ismail, F. A.; Mohd Aris, S. Towards Good Governance of Premarital Course for Muslims in Malaysia. Int Rev Manag Mark. 2016, 6(S8), 8-12. 15. Shuib, N.; Misheila, N.; Zabarani, N.; Shuib, J. Implementation of sexuality education for students with special needs (Learning disabilities). IJEAP. 2020, 379-390. 16. Fazli, K. Z.; Low, W. Y.; Merghati-Khoei, E.; Ghorbani, B. Sexuality education in Malaysia: perceived issues and barriers by professionals. Asia Pac J Public Health. 2014, 26 (4),358-366.
27. Diah, N. M.; Samsudin, S. In What, When and How to Learn About Sex: The Narratives of Students With, 1st Progress in Social Science, Humanities and Education Research Symposium (PSSHERS 2019), Atlantis Press: 2020; pp 533-537. 28. Viknesh, S. Special needs children need sex education too. TheStar 2021.
\ |
|
Second, it would be very important for your readers to understand sex education in terms of the Islamic doctrines and religious traditions, which can be used to highlight the need for sex education among children who are intellectually disabled.
|
(Page 2) Islamic teachers teach SE when the children approaching teenage years, mainly on menstruation issues, dress code, and avoiding immoral activities or illegal conducts topics as deemed by the law of the country and social norm of the community. When the person approaching end of teenage years or prior to marriage, roles and rights of married couple are taught in a formal class in school or workshop classes (Pre-marriage courses) [13]. This is made possible by the law that requires all Muslims applying for marriage in Malaysia to have the certificate of having completing this course [14].
|
|
Third, the theoretical framework was appropriate but misplaced. Instead of introducing the theoretical framework in the discussion section, I would develop a literature review section aft the Introduction. In so doing, this theoretical framework can be used as a guidance for data analysis and directions for future research. |
Line in Table 1 has been amended. I also add visible lines in the Table 2.
Theoretical framework was changed to introduction part (Page 3) |
|
In the discussion and conclusion sections, I would like to see more detailed recommendations in terms of possible evidence-based intervention strategies and programs that are appropriate for sex education provided for adolescents with IDs in Malaysia. How would such programs affect and be affected by public policies in an Islamic society?
|
In the discussion:
Since misunderstanding of religious teaching and cultural expectation has been recognized as one of main reason that discourage Muslim community to learn about SE [47], Ministry for Religion Affair and Islamic Departments in all states are invited to involve directly in the policy making and module pertaining to SE programme. This move is to incorporated more of the religious view and cultural norm in the SE programme and in doing it will reduce the community and parental barrier in accepting SE programme at school. Islamic scholars are highly looked upon and have the capability to correct societal misconception about SE that has been perceived as a culprit in encouraging teenagers to be involved in prohibited Islamic behaviors or sexual act rather than preventing them from negative impact of such behaviors [47]. The Islamic scholars also has a role in helping Muslim parents to be aware regarding the importance of SE in Islamic teachings and parental ignorance may leads to sexual harassment, premarital sexual relationships, unwanted pregnancy and lots more especially in children with ID.
In the conclusion: It should be a collaborative effort including stakeholders, healthcare practitioners, school educators, Islamic scholars and social care services to overcome the obstacles and support parents in delivering SE to their children with IDs. |
Reviewer 2 Report
- In terms of ethical concerns, the ethical issues are not described very fully. This may be because the institution did not require a more thorough discussion, but that is so, the authors should still spend some time on ethical considerations, especially given the subject under question, and with several vulnerable participants.
- In discussing the rigour of the research, which is qualitative, the authors should not refer to quantitative criteria such as validity, but instead discuss resonance, credibility, and so on. As a reader I got the sense that at least one team member was uncomfortable with the paradigm.
- The authors might have spent more time on the differences among participants' level of education and its implications for the outcomes. There is considerable literature in this area. Also, as a reader from a different cultural background I would have been very interested in a more fulsome discussion of the issues of family and cultural values related to discussions of sexuality in general, and more specifically with children with "disabilities". I was also curious that illnesses such as ADHD were included in the definition of mental disability - is this also a cultural issue?
- The use of English was quite acceptable, except for the odd interesting turn of phrase. I wouldn't recommend extensive copy-editing.
Author Response
2 |
REVIEWER 2 In terms of ethical concerns, the ethical issues are not described very fully. This may be because the institution did not require a more thorough discussion, but that is so, the authors should still spend some time on ethical considerations, especially given the subject under question, and with several vulnerable participants.
|
Thank you for comments and recommendations. I have added the ethical consideration in the text
Methodology section: 2.3 In procedure (page 5-6), we add:
While waiting for ethical permission from the The Human Research Ethics Committee of Universiti Sains Malaysia (USM), researchers identify a centre or organization that has an adolescent with intellectual disabilities. Following permission by The Study Human Research Ethics Committee, the researchers wrote a letter to the respected centre or organization to seek permission to conduct research. After obtaining the approval from the center's director or authorization, participants were identified based on inclusion criteria. Researchers received the contact number of the parents of an adolescent with disabilities from the teacher, manager, or other relevant person and scheduled face-to-face interview with the parents. Parents who understand the research objectives and volunteer to participate in the interview were selected and was given participant information sheet that contain a details of research information. The parents were also informed that some of the questions may have sensitive words/phrases that may be disturbing and they have the right to refuse to answer any questions asked by the researcher if they think the questions are too sensitive and uncomfortable. If they are unable to proceed with the interview session due to emotional instability, the interview will be stopped and will be rearranged to another time if needed. If the parents showed any signs of depression or persistent emotional instability/psychosocial problems during the interview, they shall be referred (with their consent) to the respective department (counseling/psychiatry) for expert management. The list of clinics was also given for their reference if needed. However, none of them having these problem during the research time frame. On the day of the interview, participants gave consent before the session began. Following that, participants were requested to complete a sociodemographic questionnaire. After completing the form, the interview session started by using a semi-structured questionnaire to guide the researcher during the interview.
|
|
In discussing the rigour of the research, which is qualitative, the authors should not refer to quantitative criteria such as validity, but instead discuss resonance, credibility, and so on. As a reader I got the sense that at least one team member was uncomfortable with the paradigm. |
Page 5 (Methoodology) In the data analysis, we add: - Several strategies were used to ensure the rigour and credibility of the data analysis - To guarantee trustworthiness, reliability, and proper coding, the research supervisors (RDM and ZMY) collaborated with the main researcher to double-check all interview transcripts' codes (NNK) to ensure confirmability. - In terms of reflexivity, none of the researchers had any prior contact with either the organization that aided data collection or any of the respondents. - The transcribed material was sent to chosen participants to be reviewed to verify the results and maximize the credibility. |
|
The authors might have spent more time on the differences among participants' level of education and its implications for the outcomes. There is considerable literature in this area. Also, as a reader from a different cultural background. I would have been very interested in a more fulsome discussion of the issues of family and cultural values related to discussions of sexuality in general, and more specifically with children with "disabilities". I was also curious that illnesses such as ADHD were included in the definition of mental disability - is this also a cultural issue? |
In introduction (page 2), we add:
Malaysia embraced a wide general definition of 'learning disabilities' for registration reasons. It is to enable this group to obtain free services and support from the government and agencies affiliated with the government. Under the Ministry of Women, Family, and Community Development, the Social Welfare Department (SWD) has developed seven categories of disabilities include speech, hearing, physical, vision, mental, learning, and multiple disabilities [8]. Individuals with learning disabilities have cognitive capacity that is not equivalent to their chronological age and also exhibit significant difficulties with their activities of daily life. These individuals included person with Down Syndrome, ADHD, autism, global developmental delay, intellectual disability, or specific learning disability such as dyslexia or slow learner. Upon registration, they will receive a ‘Person with disability (PWD)’ card that contained their personal information and the type of the disability. There is some stigma associated with ‘PWD’ card holders in some circumstances where they are treated as underserved and marginalized by the community [8].
In discussion, we add:
- This research discovered that parents' level of sexual knowledge and attitudes concerning SE were inextricably linked to their educational level. In general, the greater the academic level of the parents, the greater the extent of sexual knowledge and the more favourable attitudes toward providing SE to their children. This is in line with study by Xin Jin among Chinese parents that parents’ educational background had a significant effect on parental sexual knowledge and attitude [34]. As a result, parents with low academic level had a negative attitude SE due to scarcity of knowledge. - Thus, increasing awareness and accessibility on sex education was a very effective strategy for improving sexual knowledge and attitudes about sex education among parents, educators, and the general public.
References added: 1. 8. Dzalani, H.; Shamsuddin, K. A review of definitions and identifications of specific learning disabilities in Malaysia and challenges in provision of services. Pertanika J Soc. 2014, 22 (1). 2. 34. Jin, X. The characteristics and relationship of parental sexual knowledge and sex education attitude to young children. Creat Educ. 2021, 12 (9), 2002-2010. |
|
The use of English was quite acceptable, except for the odd interesting turn of phrase. I wouldn't recommend extensive copy-editing |
Thank you for the advices and observation. We highly appreciate it. The has this to Enago previously. |
Round 2
Reviewer 1 Report
The authors revised the manuscript based on the reviewers' comments and suggestions. Since this topic is understudied in an Islamic country such as Malaysia, this paper can make a solid contribution to the fields of sex education and intellectual disability studies. I recommend a thorough English line-editing before publication.